



# Saharan dust linked to European hail events

Killian P. Brennan[1] and Lena Wilhelm[2]

[1]Institute for Atmospheric and Climate Science, ETH Zürich, Zurich, Switzerland
[2]Institute of Geography – Oeschger Centre for Climate Change Research, University of Bern, Switzerland

**Correspondence:** Killian P. Brennan (killian.brennan@env.ethz.ch)

**Abstract.** Saharan dust significantly influences hail occurrence in Europe. Using Copernicus Atmosphere Monitoring Service (CAMS) and reanalysis data, crowd-sourced hail reports, lightning data, and radar measurements, we find a strong correlation between elevated dust loading and hail events. Hail coverage exceeding 28% of $1° \times 1°$ grid cells only occurs when dust loading surpasses $2.4\,\mathrm{mg\,m^{-2}}$, while on hail days the median dust load is 1.82 times higher than on non-hail days ($7\sigma$ difference). This effect is particularly strong along the Alpine crest, central France, eastern Germany, Austria, and Eastern Europe, where median dust loads more than double on hail days.

By grouping data according to synoptic weather patterns, we confirm that hail days consistently exhibit higher dust concentrations regardless of prevailing synoptic conditions, supporting the robust link between dust and hail. Peak hail activity occurs at $38\,\mathrm{mg\,m^{-2}}$ or a dust optical depth of 0.033, suggesting enhanced cloud and ice nucleation. Above this range, hail frequency declines, likely due to microphysical or radiative constraints.

Crowd-sourced reports show significantly more hail events on high-dust days, with up to 10 times more reports for hail $>20\,\mathrm{mm}$. Statistical hail models, including a logistic regression model (LRM) and a generalized additive model (GAM), rank dust as one of the top three predictors. Its inclusion increases the critical success index (CSI) by 5% (LRM) and 12% (GAM), and boosts explained variance in the GAM by 6%. These findings identify Saharan dust as a key modulator of European hail activity.

## 1 Introduction

Hail is among the most damaging atmospheric phenomena in mid-latitudes, causing extensive harm to agriculture, infrastructure, and vehicles (Changnon, 1999; Crompton and McAneney, 2008). Understanding the processes leading to hail formation is crucial for improving predictions and mitigating severe weather impacts in Europe.

Saharan dust events, characterized by the transport of mineral dust from the Sahara Desert across the Mediterranean into Europe, are the predominant source of atmospheric dust loads in the region over the past 40 years (Varga, 2020; Brunner et al., 2021). These dust plumes contribute significantly to European aerosol concentrations, affecting weather patterns and precipitation (Masson et al., 2010; Rodríguez et al., 2001). Subtropical anticyclones shifting to higher latitudes and amplified Rossby waves are associated with extreme Saharan dust events (Rodríguez and López-Darias, 2024). Saharan dust plays a crucial role in cloud formation by serving as both cloud condensation nuclei (CCN) and ice-nucleating particles (INP) (Boose et al., 2016; Twohy et al., 2009; Meloni et al., 2008).





CCNs influence cloud droplet concentrations and size distributions, affecting precipitation initiation and hail formation. Higher CCN concentrations lead to numerous smaller droplets, suppressing coalescence and delaying precipitation onset (Rosenfeld et al., 2008). This delay allows more supercooled liquid water to be carried aloft, enhancing hailstone growth through accretion (Khain et al., 2005). Increased CCN levels can also modify latent heat release within clouds, strengthening updrafts and potentially intensifying thunderstorm dynamics (Tao et al., 2012).


INPs promote ice formation at higher temperatures, facilitating the early initiation of ice processes within clouds (Boose et al., 2016). Variations in INP concentrations affect initiation of freezing, influencing hailstone size and distribution. Higher INP concentrations increase ice crystal numbers, leading to numerous smaller hailstones, while lower concentrations may result in fewer but larger hailstones due to more supercooled liquid water per nucleus (Cantrell and Heymsfield, 2005; DeMott et al., 2010).


Beyond their microphysical roles, Saharan dust particles impact the Earth's radiation budget through radiative effects. Dust aerosols scatter and absorb solar and terrestrial radiation, modifying atmospheric heating rates and surface temperatures (Osborne et al., 2011). This atmospheric heating can alter stability profiles and affect convection, potentially suppressing or enhancing it depending on environmental conditions (Perlwitz et al., 2001). The semi-direct effect, where absorbing aerosols heat the atmosphere and change relative humidity profiles, influences cloud development and lifetime (Hansen et al., 1997).


Observational studies have provided valuable insights into the role of aerosols in cloud behavior and precipitation processes. A study of shipping lane emissions demonstrated a 40% decline in lightning linked to reduced aerosol pollution, highlighting the significant impact of aerosols on deep convective cloud behavior and aerosol–cloud–interactions (Wright et al., 2024). Similarly, recent research has identified Saharan dust as a significant factor influencing rainfall in tropical cyclones. Utilizing machine learning models with extensive meteorological data and satellite observations, Zhu et al. (2024) found that dust is a key predictor for tropical cyclone rainfall, revealing a nonlinear relationship where rainfall increases with dust concentration up to a certain point (dust optical depth of 0.06) before decreasing sharply at high dust levels. This finding underscores the complex role of Saharan dust in precipitation processes and the importance of accounting for nonlinear effects.


Building on these observations, modeling studies have explored how variations in CCN and INP concentrations affect cloud properties, precipitation outcomes, and hail characteristics. In single-cell convection events with low convective available potential energy (CAPE) and minimal wind shear, increased CCN concentrations tend to decrease total surface precipitation and peak updraft velocities (Seifert and Beheng, 2006; Lee et al., 2008a). Conversely, in multi-cell storms characterized by higher instability and shear, higher aerosol levels can enhance surface precipitation due to increased evaporation, stronger updrafts and downdrafts, stimulation of new convective cells, and improved convective organization (Khain et al., 2005; Lynn et al., 2005; Seifert and Beheng, 2006; Lee et al., 2008b). In Alpine environments, aerosol–cloud–precipitation interactions have been investigated through case studies that underscore the role of aerosol concentrations. Eirund et al. (2022) found that forecast models lacking aerosol perturbations underestimated heavy precipitation leading to flooding in northeastern Switzerland; sensitivity simulations showed that increased CCN and INP concentrations significantly impacted convective dynamics and precipitation patterns. Barthlott et al. (2024) observed that only high-resolution models with low CCN concentrations accurately captured certain thunderstorms, underscoring the significant role of aerosol concentrations in simulations. Loftus and








Cotton (2014) indicate that increasing CCN concentrations generally result in larger and more numerous hailstones, even if overall storm dynamics remain largely unchanged. Hydrometeor responses and hail formation exhibit non-monotonic behavior with varying CCN levels, emphasizing the need for further research across different storm environments. For instance, Ilotoviz et al. (2018) examined the relationship between aerosols and hail two-dimensional cloud model with spectral bin microphysics, finding that high aerosol concentrations enhance supercooled cloud water content and promote wet hail growth, resulting in larger hailstones.

Despite these findings, the specific influence of Saharan dust on hail occurrence in Europe remains underexplored. Given Saharan dust's significant role as a source of CCNs and INPs and its impact on atmospheric processes, our study aims to address the following research questions:

1. What effect do high and low aerosol dust mass concentrations have on the occurrence of hail in Europe?

2. Can aerosol dust loading be used as a predictor to improve statistical hail prediction models?

To answer these questions, we examine historical data from sources including the ECMWF CAMS model (Inness et al., 2019, Sect. 2.1), radar observations (Sect. 2.2.1) and (Sect. 2.2.2), lightning measurements (Sect. 2.3), crowd-sourced reports (Sect. 2.2.3), and ERA5 reanalysis data (Sect. 2.4). We aim to uncover patterns and correlations between dust events and hail formation across Europe (Sect. 3). Statistical models are utilized to quantify the statistical relationship between hail day occurrence and dust concentrations, enabling us to further infer the potential influence of dust on hail formation processes and to address research question 2 (Sect. 4). Conclusions are drawn and future research avenues are suggested in Sect. 5.

## 2 Data and methods

In this study, we combined reanalysis data, crowd-sourced reports, lightning, and radar measurements to investigate the influence of Saharan dust events on hail formation in Europe. In order to investigate the effect of altered dust loads on hail occurrence and not thunderstorm occurrence in general, only local days with lightning were included in the analysis performed in this study (Sect. 2.3). Further, as most hail occurrences in Europe are constrained to the summer season, only the peak hail months of June, July, and August were considered in this analysis. With the exception of the analysis incorporating the crowd-sourced reports (Sect. 2.2.3 and the statistical models (Sect. 4) the analysis was conducted for the period 2013 – 2021 (828 days). The analysis was conducted on a $1° \times 1°$ grid. Applying the hail area fraction (Sect. 2.2.1) and lightning density thresholds (Sect. 2.3), from the total 385 390 grid-point days, 86 900 are grid-point thunderstorm days of which 32 989 are local, grid-point hail days (38% of grid-point thunderstorm days). The analysis presented herein was conducted from 10°W to 25°E & 35°N to 53°N.

## 2.1 Dust data

To determine whether the target regions were influenced by Saharan dust, we used the total vertically integrated aerosol dust from the Copernicus Atmosphere Monitoring Service (CAMS, Inness et al., 2019). The CAMS reanalysis is a global aerosol





dataset available from 2003 onwards with annual updates, produced by ECMWF. It assimilates satellite dust optical depth data using the Integrated Forecasting System, offering improved spatial resolution ($80\,\text{km}$) and finer temporal resolution over previous versions. With reduced biases and enhanced consistency, it serves as a valuable resource for climatology, trend analysis, model evaluation, and providing boundary conditions for regional models. The dust data from CAMS provides a quantitative measure of the dust concentration in the atmosphere, allowing us to identify periods and areas affected by Saharan dust transport. Specifically, we used vertically integrated mass of dust aerosol in three size ranges $0.03 - 0.55$, $0.55 - 9$, and $9 - 20\,\mu\text{m}$. This dataset has a horizontal resolution of $1° \times 1°$, and all other data presented in the following will be brought onto this grid for analysis. We used daily 15 UTC timesteps for our analysis, as this represents the maximum in the hail diurnal cycle in Europe (Cui et al., 2024). In this text *dust loading* is used as a short form for "vertically integrated mass of dust aerosol".

## 2.2 Hail observations

### 2.2.1 EURADHAIL

To determine hail events, we use the European radar hail climatology (EURADHAIL Cui et al., 2024), which is based on OPERA (Huuskonen et al., 2014). Following Cui et al. (2024), a reflectivity threshold of $53\,\text{dBZ}$ classifies hail in the EURAD-HAIL dataset. From this, high resolution (2 km) daily ($06 - 06$ UTC) binary fields of hail occurrence are calculated. Further, to reduce the spatial resolution of this dataset, the fraction of area covered by hail is computed for each $1° \times 1°$ grid-point if OPERA data is available for at least 50% of the grid-point area. This quantity is referred to as the hail area fraction and is available for 140 grid-points. A threshold of 0.01 for the hail area fraction was used to classify a given grid-point as a local hail day, this is equivalent to the threshold used in Barras et al. (2021), and Wilhelm et al. (2024). This dataset encompasses the period $2013 - 2021$.

### 2.2.2 POH

For statistical models analyzing the effect of dust loading on hail day occurrence, we focus on northern Switzerland, using hail day data from the Probability of Hail (POH) radar product (Germann et al., 2022). POH is an empirical hail detection algorithm estimating ground-level hail probability ($0 - 100\%$) based on the vertical distance between the 45 dBZ echo top height and the freezing level height, following Waldvogel et al. (1979). This approach is more accurate in capturing hail events than EURADHAIL, since it does not include the freezing level.

Daily POH data from 2003 to 2022 were analyzed for the region north of the Swiss Alps and the foothills (see Wilhelm et al., 2024). Hail days were defined using thresholds adapted from Barras et al. (2021), with a day classified as a hail day if more than $580\,\text{km}^2$ had POH $>$80%.

### 2.2.3 Crowd-sourced hail reports

Swiss crowd-sourced hail reports as detailed in Barras et al. (2019) provide valuable, localized observations of hail events, contributing to the spatial and temporal mapping of hail occurrences across the target region. Since May 2015, more than





300 000 hail size reports have been gathered across Switzerland via the MeteoSwiss app, which allows users to report the time, location, and size of hailstones they observe. The discrete size categories range from "no hail" to "tennis ball" size, with users able to manually adjust the time and location of their reports. These reports undergo a multi-step plausibility check, including spatial and temporal validation against radar data, to filter out erroneous entries. The crowd-sourced data provides extensive spatial coverage and valuable ground truth for hail events, complementing radar-based measurements. In order to account for the growth in user base over time, the fraction of reports made on a given day in relation to the total reports in that year was calculated for each size bin. The crowd-sourced hail reports collected in the years 2015 – 2021 were used in this study.

## 2.3 Lightning measurements

Daily lightning data was retrieved from the World Wide Lightning Location Network (WWLLN) Global Lightning Climatology (Kaplan and Lau, 2022). The mean lightning density within each $1° \times 1°$ grid-point was calculated, and for each grid-point, days with a mean lightning density greater than $0.0005\,\mathrm{km^{-2}\,d^{-1}}$ were considered to be thunderstorm days.

## 2.4 Reanalysis data

We further employ the ERA5 reanalysis dataset, the latest product from the European Centre for Medium-Range Weather Forecasts (ECMWF), representing the fifth generation of atmospheric reanalysis data (Hersbach et al., 2020). This dataset offers a spatial resolution of $31\,\mathrm{km}$ (spectral resolution T369) for data on single levels, pressure levels, and model levels (137 vertical levels) with data available at an hourly temporal resolution. For the synoptic clustering analysis the ERA5 geopotential height at $500\,\mathrm{hPa}$ pressure level 15 UTC was used. The environmental predictors for the statistical model were calculated from ERA5 model level, pressure level, and single-level data for a study region in northern Switzerland, spanning 2003-2022 in hourly resolution. A total of 75 convective parameters were calculated (see Wilhelm et al., 2024).

## 3 Co-occurrence of dust and hail

In this section, we investigate the local statistical dependencies between aerosol dust loading with the occurrence of hail.

Comparing thunderstorm days with the lowest and highest dust loading reveals that especially high hail area fraction values are predominantly found during periods of high dust loading and vice-versa (Fig. 1). Hail area fractions larger than 0.28 were found exclusively during periods with a dust loading larger than $2.4\,\mathrm{mg\,m^{-2}}$. This threshold is close to the median climatological dust concentration across the EURADHAIL domain (54[th] percentile).

In fact, the median dust load during local hail days is 1.82 times higher than during non-hail days. The difference in mean dust loading between local hail days and non-hail days is highly significant ($7\sigma$, Welch's t-test) and valid ($>3\sigma$) for a wide range of local hail day area thresholds (0.006 – 0.42, as defined in Sect. 2.2.1). This relationship is present throughout Europe, however, most of the significant areas lie within the main Alpine crest, central France, eastern Germany, Austria, and much of eastern Europe with wide-spread areas where the median dust load during local hail days is more than twice as high than during non-hail days (Fig. 2).





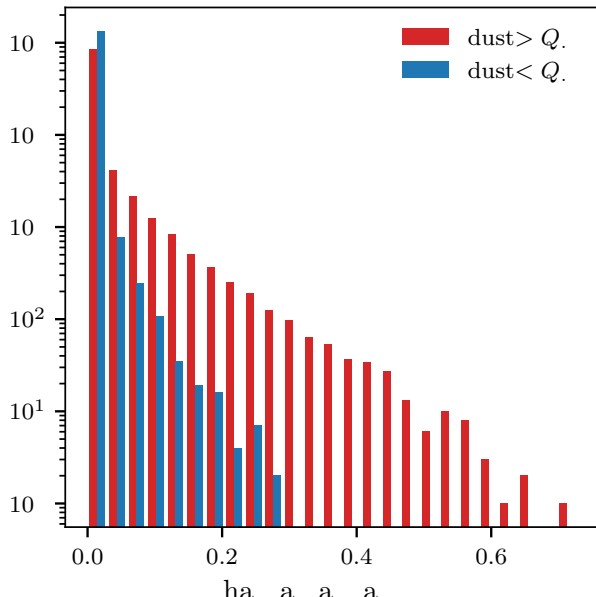

**Figure 1.** Histogram of hail area fractions of thunderstorm days with lowest and highest 30 percentile vertically integrated mass of dust aerosol $0.03 - 20\,\mu m$ (CAMS, 2.4 and $22\,\mathrm{mg\,m^{-2}}$, blue and red respectively).

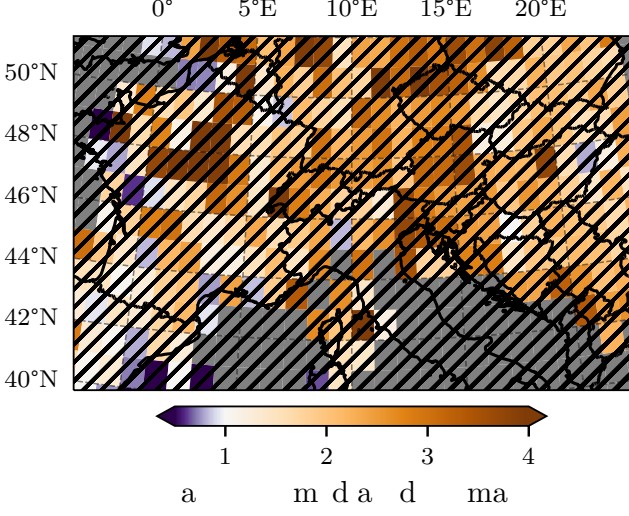

**Figure 2.** Gridpoints colored by their factor of median vertically integrated mass of dust aerosol $0.03 - 20\,\mu m$ (CAMS) during local hail days versus non-hail days (EURADHAIL). Grid-points with less than three hail days per year are greyed out, while areas with $< 1\sigma$ significance are hatched.



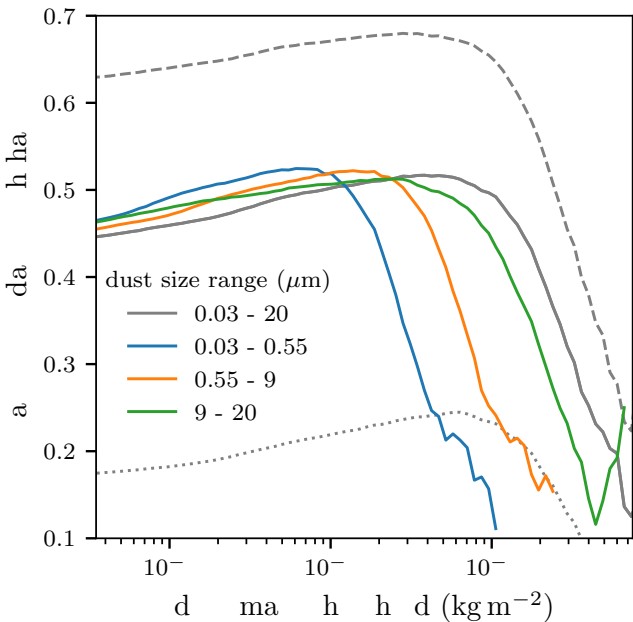

**Figure 3.** Fraction of days with hail (EURADHAIL) as a function of vertically integrated mass of dust aerosol (CAMS) for different dust size bins (colors) and different hail area fraction thresholds (dotted, solid, and dashed lines represent 0.002, 0.01, and 0.05 area fractions respectively). Values for dust mass that occur on less than 10 grid-point days were omitted.

There exists a distinct maxima in the fraction of grid-point days with hail occurrence as a function of dust mass (Fig. 3). Depending on the dust size range considered and hail area fraction used to define local hail days (Sect. 2.2.1), this maxima occurs at 6.1, 13.8, and $25.4\,\mathrm{mg\,m^{-2}}$ for the aerosol size ranges $0.03 - 0.55$, $0.55 - 9$, and $9 - 20\,\mu\mathrm{m}$ respectively, and in the size range $0.03 - 20\,\mu\mathrm{m}$ at 28.1, 38.2 and $63.6\,\mathrm{mg\,m^{-2}}$ for the hail area fraction thresholds 0.002, 0.01 and 0.05 respectively. At concentrations lower than said peak, there is an exponential relationship between dust loading and local hail day fraction and

a steep decline in local hail day fraction after the maximum. A similar nonlinear relationship between dust mass and tropical cyclone rainfall was discovered by Zhu et al. (2024), who also found a decrease in values beyond the peak response. The peak response at $38\,\mathrm{mg\,m^{-2}}$ likely hints towards a beneficiary microphysical effect of the aerosols at dust concentrations below $60\,\mathrm{mg\,m^{-2}}$ and a limiting effect due to other microphysical or radiative processes at dust concentrations higher than that. The peak of $38\,\mathrm{mg\,m^{-2}}$ falls on the 84[th] percentile of the dust climatology. To relate our findings with Zhu et al. (2024), who found

an intermediate peak at a dust optical depth of 0.06, we can convert our vertically integrated aerosol dust mass numbers to dust optical depth by multiplying with a factor of $785\,\mathrm{kg\,m^{-2}}$ (correlation between vertically integrated aerosol dust mass $0.03 - 20\,\mu\mathrm{m}$ and dust optical depth is 0.997). Using this conversion, our intermediate maxima lies at 0.033, which was verified through another approach (Fig. A1).



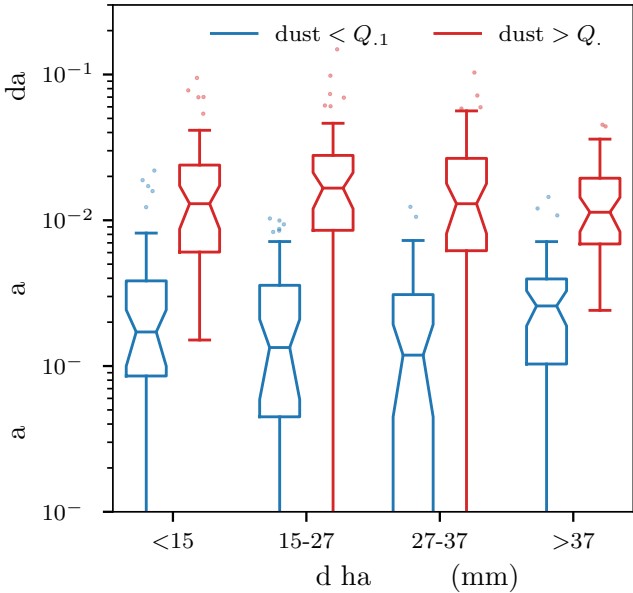

**Figure 4.** Fraction of yearly crowd-sourced reports that were submitted in Switzerland on a given thunderstorm day for different reported hail size categories during the 10% highest and 10% lowest dust load days (CAMS mean vertically integrated mass of dust aerosol $0.03 - 20\,\mu\text{m}$ in the area $5°\text{E}$ to $11°\text{E}$ & $45°\text{N}$ to $48°\text{N}$, red and blue respectively). 43 days fall into each category. Only thunderstorm days with South to West flow are shown (mean $700\,\text{hPa}$ flow $>1\,\text{m s}^{-1}$, ERA5 reanalysis (Hersbach et al., 2020)). Horizontal line with notches shows the mean and confidence interval respectively, while the box covers the interquartile range (IQR) and the whiskers extend to the farthest data point lying within $1.5\times$ the IQR from the box.

## 3.1 Effect on hail size

When comparing crowd-reported hail sizes in Switzerland during the lowest and highest dust load days, it is evident, that regardless of size category, significantly ($>4\sigma$) more hail reports are submitted during high dust days (Fig. 4). In fact, on average, 5.2, 9.8, 10.3, and 4.7 times more reports are submitted during high dust days ($90^{\text{th}}$ percentile, $85\,\text{mg m}^{-2}$) versus low dust days ($10^{\text{th}}$ percentile, $2.6\,\text{mg m}^{-2}$) for hail size categories of $<15$, $15-27$, $27-37$, and $>37$ mm respectively. With exception of the largest hail size category, the difference in the reports submitted during high vs low dust load days increases

with larger hail sizes. The nonlinearity observed in the largest crowd-sourced hail size category could be associated to false reports. As discussed in Kopp et al. (2024) and references therein, the largest hail categories exhibit the highest number of false reports.



## 3.2 Isolating synoptic influence

In an attempt to further rule out synoptic effects as a hidden confounder in the relationship between hail and dust, we conducted
the analysis presented in Section 3 on days with similar synoptic situations. To this end, using principal component analysis
(PCA, Pedregosa et al., 2011) with the explained variance set to $1\sigma$, yielding four dimensions, and K-means clustering (Krishna
and Narasimha Murty, 1999) we grouped the 15 UTC geopotential height $Z$ at $p = 500\,\text{hPa}$ in a $14° \times 20°$ box (coarsened to
$7 \times 7$ pixels) centered on the target region 7°E to 12°E & 47°N 49°N. In the selected target region, $n = 254$ lightning days were
assigned to 11 clusters, a choice that achieved a maximum Silhouette score of $0.25$ (Rousseeuw, 1987). Within each cluster, the
dust load in each target gridpoint was compared between hail and non-hail days. It was found that the most populous cluster
with $n = 44$ had in the median, 1.86 times more dust during hail days $(1.6\sigma)$, while the next three most populous clusters
$(n = 30$ to $34)$ yielded dust mass factors of 1.77 $(1.5\sigma)$, 0.81 $(0.13\sigma)$, and 5.46 $(4.8\sigma)$. The mean dust factor across all clusters
weighted by $n$ was 3.20 $(1.6\sigma)$.

This analysis was repeated for a different target regions (13°E to 16°E & 47°N to 49°N, 15°E to 17°E & 44°N to 46°N,
and 0°E to 4°E & 47°N to 49°N) with $n = 272$, $n = 263$, and $n = 97$ lightning days respectively. In these target regions, 9, 10,
and 13 clusters respectively performed best (Silhouette scores of 0.27, 0.28, and 0.27) and weighted mean dust mass factors of
2.70 $(2.1\sigma)$, 3.27 $(1.5\sigma)$, and 3.42 $(1.1\sigma)$ were determined.

The geopotential height clustering analysis confirms the robustness of the observed relationship between dust and hail. By
grouping days with similar synoptic conditions using PCA and K-means clustering, distinct clusters were identified, allowing
for targeted comparisons of dust load between hail and non-hail days. Across all tested regions and clustering approaches,
the weighted mean dust mass factor consistently exceeded 1, reinforcing the conclusion that increased dust concentrations are
associated with hail days. The results remained stable across variations in clustering methods (such as self-organizing maps or
hierarchical clustering) and parameters, further ruling out synoptic effects as a confounding factor.

Differences in the thermodynamic environments across the categories—lightning, hail, dust, hail with dust, and normal
days—are detailed in the Appendix.B.

## 4   Statistical model

Building on the observed positive relationship between hail days and high dust loading, this section investigates how statistical
models capture and quantify the relationship between hail occurrence and dust loading and, more importantly, whether the
inclusion of dust predictors enhances predictive performance.

To this end, we employed a logistic regression model (LRM) and a generalized additive model (GAM) to predict hail events
in northern Switzerland, a hail hotspot in Europe. Logistic regression is widely used in atmospheric sciences to predict binary
outcomes such as hail or thunderstorm occurrence (e.g., Wilhelm et al., 2024; Battaglioli et al., 2023). Here, we used it to
estimate the probability $p(x)$ of a hail day from POH (2.2.2) based on environmental predictors from ERA5 (2.4) and dust



loading from CAMS (2.1). The logistic regression model is expressed as:

$$p(x) = \frac{1}{1 + e^{-g(x)}}, \quad \text{where} \quad g(x) = \beta_0 + \beta_1 x_1 + \beta_2 x_2 + \cdots + \beta_n x_n. \tag{1}$$

where, $x_i$ represent environmental predictors and $\beta_i$ are regression coefficients determined via maximum likelihood estimation using the glm package in R. To prevent overfitting, the dataset (2013–2022) was divided into training (60%), testing (20%) and validation (20%) sets. Ten-fold cross-validation was used to evaluate model performance based on multiple metrics (1).

Generalized additive models (GAMs) were implemented to account for potential nonlinear relationships between predictors and hail occurrence. GAMs extend logistic regression by incorporating spline-based smooth functions $f_i(x_i)$:

$$g(x) = \beta_0 + f_1(x_1) + f_2(x_2) + \ldots + f_n(x_n) \tag{2}$$

enabling the capture of non-monotonic effects, such as those of dust loading. The GAMs were implemented using the mgcv package in R, with smoothing parameters optimized via generalized cross-validation.

The goal was to build a physically informed, ingredient-based model incorporating predictors relevant to hail formation: atmospheric instability, moisture, and wind shear. Unlike prior studies optimizing predictive performance (e.g., Wilhelm et al., 2024; Battaglioli et al., 2023), the focus here was on simple parameters to evaluate the impact of including dust as a predictor.

Predictors in both models included the sum of the vertically integrated mass of dust aerosol across three size ranges (0.03 – 0.55, 0.55 – 9, and 9 – 20 $\mu$m), the convective available potential energy (CAPE), capturing atmospheric instability, the wind shear magnitude from the surface to 500 hPa ($WS$), important for storm organization, and the height of the freezing level ($z\_0°C$) which influences the depth of the hail growth zone. For moisture availability, the LRM used 2 m dewpoint temperature ($Td_{2m}$), while the GAM included mean relative humidity across 850–500 hPa ($RHmid$). In both models, the dust predictor was highly significant, with GAMs showing high degrees of freedom, indicating a nonlinear relationship with hail occurrence.

CAPE was the most important predictor in both models, followed by dust and/or moisture, as shown by $z$-values in the LRM and Chi-squared/SHAP values in the GAM. This highlights the relevance of dust-related predictors in statistical hail modeling, though their importance varies with model type, parameter tuning, choice of covariates and data resolution.

Both models performed well, achieving AUROCs of 0.88 (LRM) and 0.91 (GAM). However, high false alarm rates (FARs) of 0.49 (LRM) and 0.44 (GAM) were observed, a common issue in extreme event classification. Including dust loading as a predictor improved the predictive skill of both models, e.g. increasing the critical success index (CSI) by 5 % in the LRM and 12 % in the GAM (1). In the GAM, explained variance rose by 6 % indicating that dust predictors contribute valuable additional information about hail day occurrence, although the overall variance explained remains low due to the complexity of hail dynamics.

The GAM reveals that hail days are more likely at intermediate dust concentrations, peaking around $12\,\mathrm{mg\,m^{-2}}$ and declining at very high dust loads at the end of the distribution (Fig. 5). Note that this value is smaller than what the analysis in Sect. 3 showed because the dust loading was averaged over the northern Swiss domain. This "boomerang" shape aligns well with findings from Zhu et al. (2024) and supports the concept of an optimal aerosol loading (Koren et al., 2008). These results suggest that Saharan dust does not inherently suppress hail formation; rather, its effects depend on the concentration.




**Table 1.** Performance metrics for the logistic regression models (LRM) and generalized additive models (GAM) with and without dust as a predictor. For AUROC, POD, CSI and HSS a value close to one is optimal, while for FAR zero is optimal.

| Metric | LRM | LRM Dust | GAM | GAM Dust |
|--------|--------|----------|--------|----------|
| AUROC  | 0.8808 | 0.8883   | 0.8837 | 0.9105   |
| POD    | 0.6603 | 0.6750   | 0.6560 | 0.6942   |
| FAR    | 0.4914 | 0.4667   | 0.4444 | 0.3778   |
| CSI    | 0.4031 | 0.4244   | 0.4302 | 0.4884   |
| HSS    | 0.5068 | 0.5303   | 0.5341 | 0.5959   |

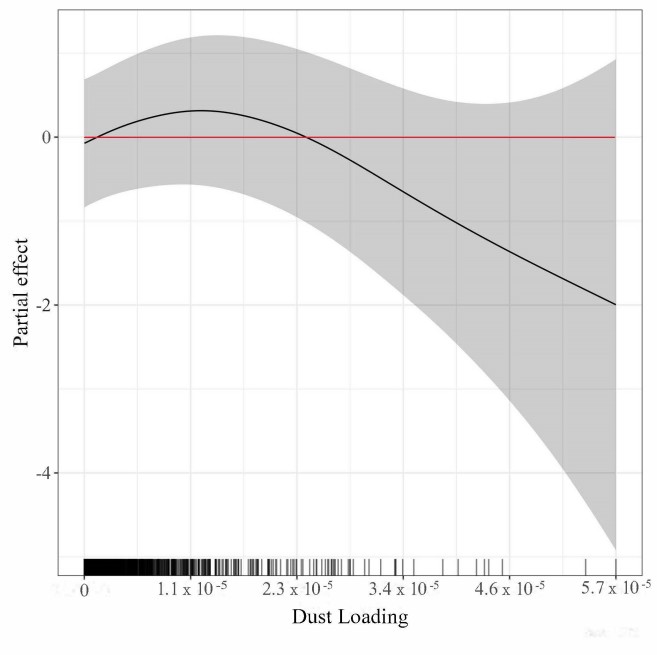

**Figure 5.** Partial dependence plots for the dust loading covariate in the GAM. The solid black line and gray uncertainty range represent the modeled partial effect of the covariate on the response. The red y 0 lines separate positive from negative effects. The short black vertical lines indicate the covariate distribution.

Moderate dust levels likely enhance microphysical processes such as ice nucleation or cloud droplet activation, while higher concentrations may inhibit hail formation through radiative or microphysical feedback. This aligns with the findings discussed in Sect. 3, reinforcing the earlier observation of dust's dual role in hail formation.





## 5 Summary and conclusions

In this study, we investigated the relationship between aerosol dust loading and hail occurrence across the EURADHAIL domain. We found that periods of high dust loading are consistently associated with the most intense hail events (Sect. 3). Specifically, the median dust load on hail days was 1.82 times greater than on non-hail days, a robust and highly significant difference ($7\sigma$) observed across multiple hail area fraction thresholds. The most extensive hail events — those covering more than 28% of $1° \times 1°$ tiles — occur exclusively on days when dust loads exceed $2.4\,\mathrm{mg\,m^{-2}}$. We also identified a distinct maximum in hail occurrence at $38\,\mathrm{mg\,m^{-2}}$, corresponding to a dust optical depth of 0.033. Importantly, our analysis of subsets of days with similar synoptic conditions through a clustering approach using geopotential height at $500\,\mathrm{hPa}$ confirmed that these patterns persist even when accounting for large-scale weather patterns (Sect. 3.2). This greatly reduces the likelihood that synoptic variability alone drives the observed association between hail days and dust loading.

Our results contribute to the evolving understanding of aerosol–cloud–hail interactions and are consistent with theoretical frameworks that describe aerosol invigoration and optimal aerosol loading. Previous studies have shown that aerosol particles acting as CCN and INP can enhance convective development and influence precipitation formation (e.g., Rosenfeld et al., 2008). The "optimal aerosol loading" concept (Koren et al., 2008) suggests that intermediate aerosol concentrations can maximize convective intensity before additional loading begins to suppress convection through radiative and microphysical feedback. This aligns with our observation of a hail occurrence peak at intermediate dust loads. Further, our findings extend work by Zhu et al. (2024), who identified similar nonlinear aerosol–precipitation relationships in tropical cyclone rainfall, into the mid-latitude convective storm regime.

Contrary to some expectations drawn from microphysical competition pathways (e.g., Sulakvelidze et al., 1974), our crowdsourced hail reports did not reveal a statistically significant effect of dust on hailstone size (Fig. 4). This suggests that while dust aerosols can enhance hail occurrence, their influence on hailstone growth processes may differ from earlier conjectures and is not hail-size dependent.

The integration of aerosol dust loading into logistic regression and GAM frameworks (Sect. 4) reveals two key findings: One, statistical modeling offers an effective approach to enhance our understanding of hail formation processes. And two, incorporating dust predictors significantly increases the predictive skill of proxy-based hail prediction models. In our simplified ingredients-based model, the dust predictor consistently emerged as a highly significant variable, ranking as the second or third most important feature. Its inclusion enhances model performance, increasing the critical success index (CSI) by 5 % in the LRM and 12 % in the GAM, while boosting explained variance in the GAM by 6 %. Incorporating aerosol variables may enable models to capture microphysical mechanisms that are typically inaccessible in large-scale environmental datasets like ERA5. Moreover, advancements in machine and deep learning offers opportunities to overcome limitations such as collinearity, allowing the use of a larger number of predictors. We recommend further investigation of dust predictors across diverse modeling frameworks. Initial experiments with random forest models and a European-scale model (not shown) have yielded promising results, demonstrating robust relationships and reliable performance when well tuned.





Incorporating aerosol variables from datasets like Inness et al. (2019) holds significant potential for improving hail forecasting models, supporting the integration of aerosol data into operational weather prediction systems. Moreover, regional weather
simulations with sophisticated microphysical parameterizations (e.g., Seifert and Beheng, 2006; Noppel et al., 2010) might benefit from explicitly representing spatial and temporal variations in dust loading. While such detailed simulations are computationally demanding, dust-related parameters could also enhance post-processing algorithms, potentially improving hail forecasts even in simpler operational models with one-moment schemes (e.g., Brennan et al., 2024).

These findings also hold implications for hail suppression efforts. Previous work on hail prevention and cloud seeding has
explored the introduction of artificial aerosols to reduce hail damage, though with limited conclusive outcomes (e.g., Atals, 1977; Federer et al., 1986; Rivera et al., 2020; Pirani et al., 2023). Our study suggests that, under certain atmospheric conditions, natural dust aerosols may actually enhance hail formation, raising the possibility that indiscriminate or poorly targeted seeding efforts could inadvertently increase hail occurrence. While caution is warranted in extrapolating these observational results directly to cloud seeding applications, our results underscore the complexity of aerosol–cloud interactions and the need for
more targeted experimental and modeling investigations before relying on aerosol-based interventions for hail mitigation.

Naturally, our correlational approach cannot fully establish causation. Although we leverage lightning data to differentiate hail-producing from non-hail-producing storms and control for synoptic conditions via clustering analyses, other confounding factors such as boundary-layer moisture or vertical wind shear could still influence the dust–hail relationship. Nonetheless, our comprehensive use of all available data and robust analytical approaches provides strong evidence that variations in dust
loading meaningfully relate to hail events. Regardless of the underlying mechanisms that lead to a link between hail and dust, the co-occurrence of hail and dust events revealed through our analysis rules out dust as a generalizable limiting factor for hail formation.

In conclusion, our study highlights a significant and robust association between aerosol dust loading and hail events across Europe. We identify a clear quantitative relationship — high hail area coverage occurs only above a dust load threshold of
$2.4\,\mathrm{mg\,m^{-2}}$, and a pronounced maximum in hail occurrence emerges at moderate dust concentrations. The complexity of these interactions, however, underscores the need for further research to clarify causal mechanisms and develop strategies that leverage aerosol information effectively.

## Appendix A:  Optimal dust optical depth

To make it easier to compare our findings with existing literature (i.e., Zhu et al., 2024), we provide a subset of our findings
using dust optical depth instead of vertically integrated dust aerosol mass (Fig. A1).

## Appendix B:  Storm environment

Following the methodology devised in Zschenderlein and Wernli (2022), the position of thunderstorm, hail, and dust days in the distribution of respective environmental variables was determined. To this end, temperature and water vapor mixing ratio at





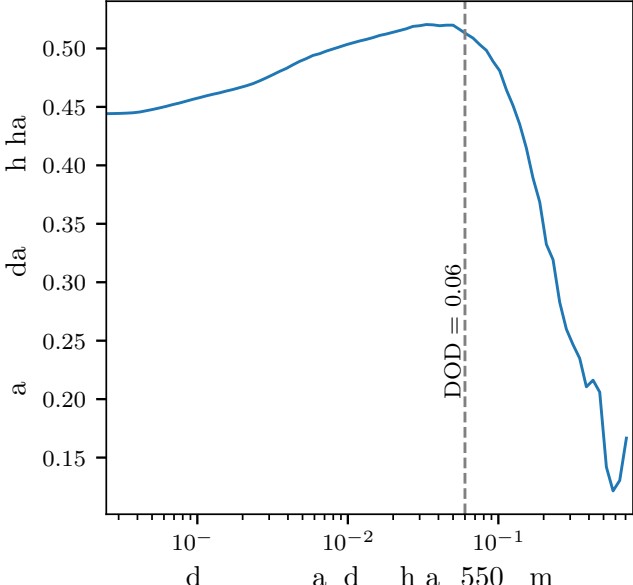

**Figure A1.** Identical to Fig. 3, but only for the 0.01 area fraction and using dust optical depth instead of vertically integrated dust aerosol mass. The dust optical depth (DOD) value of maximum response found in Zhu et al. (2024) is indicated with a dashed vertical line.

inflow height ($850\,\mathrm{hPa}$), as well as CAPE was determined. This analysis reveals, that dust days lie predominantly at the upper

end of the $T_{850\,hPa}$ distribution of all days (Fig. B1a). In comparison, lightning and hail days follow the all days distribution more closely, while exhibiting less spread. Concerning low-level humidity, the pattern is flipped, where dust days are dryer compared to the distribution of all days, and lightning and hail days are at the moister end of the distribution (Fig. B1b). Next, distributions in vertical wind shear are similar between dust and hail days with dust and differ from the climatology (Fig. B1c). Finally, the distribution of CAPE found during dust days is skewed relative to the CAPE distribution of all days, with higher

CAPE values being relatively more frequent during dust days. Surprisingly, the CAPE distribution for lightning and hail days follows more closely that of the all-day distribution (Fig. B1d).

     Our analysis of the synoptic environment reveals that although local dust days fall on the warmer side of the low-level temperature distribution and exhibit a distribution of CAPE skewed towards higher values compared to hail days, dust days also have less low-level moisture than hail days.

*Author contributions.* **KPB**: Conceptualisation; investigation; visualisation; writing — original draft, review and editing. **LW**: Statistical modelling, visualisation; writing — original draft, review and editing



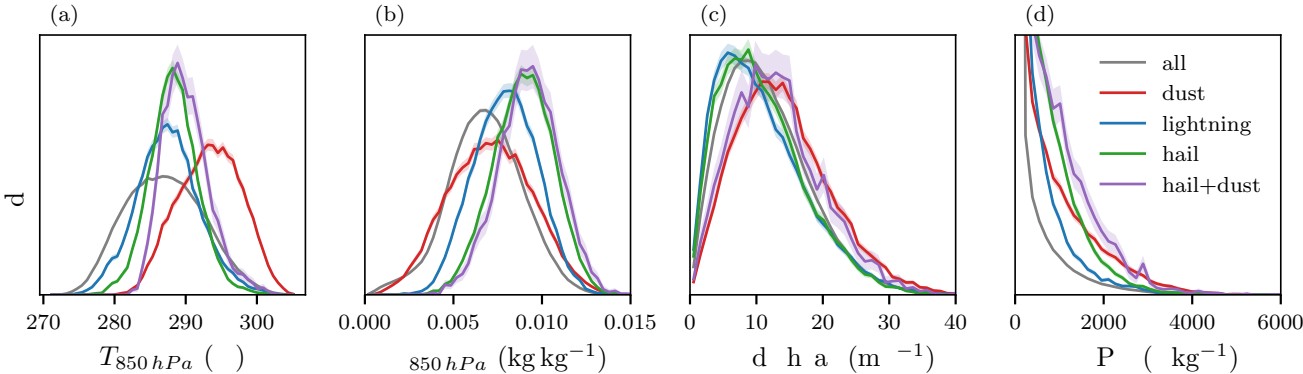

**Figure B1.** Normalized probability density of local temperature (a), and specific humidity (b) at $850\,\mathrm{hPa}$, (c) $10\,\mathrm{m}$ to $500\,\mathrm{hPa}$ vertical wind shear, and CAPE (d) at 15 UTC for each grid-point within the study domain, for all days (JJA, $2013 - 2023$), dust days, lightning days, hail days, and combined hail and dust days (ERA5). Solid lines show the distribution, while shaded areas indicate the 95% confidence interval determined through bootstrapping ($n = 1000$).

*Competing interests.* The Authors declare no conflict of interest.

*Acknowledgements.* We thank Anna Miller, Heini Wernli, Marc Federer, and Emmanouil Flaounas for their valuable input. Further, we would like to express our sincere gratitude to our colleagues from ETH and the University of Bern and the entire scClim team (https: //scclim.ethz.ch/) for their invaluable insights and discussions. This study was funded by the Swiss National Science Foundation (SNSF) Sinergia grant CRSII5_201792. We acknowledge the use of OpenAI's GPT in assisting with language refinement in the preparation of this manuscript.



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
