# Peer review of "Saharan dust linked to European hail events"

_EGUsphere, 2024_

## Author Comment (AC2)

**Author's comments for paper egusphere-2024-3924**

**Saharan dust linked to European hail events**

by Killian P. Brennan and Lena Wilhelm

June 26, 2025
* * *
**Reviewer 1**

**Summary**

Dust aerosols have an important influence on cloud formation and development. This manuscript analyzes the influence of Saharan dust on hail in Europe, which has important scientific significance. Nevertheless, the manuscript leaves much to be desired. Here are some specific comments:

**Reply**:  We thank the reviewer and appreciate your recognition of the scientific significance of our study on the influence of Saharan dust on hail in Europe. We also acknowledge your concerns regarding areas where the manuscript can be improved. Below, we provide detailed responses to your specific comments and outline the revisions we have made to enhance the clarity, rigor, and completeness of our study.

**Specific and technical comments**

**Reviewer Comment 1.1**  —  Lines 43-44: It's inappropriate to cite unpublished papers.

**Reply 1.1**:  We respectfully disagree, as staying up to date with the latest research is essential for providing a state-of-the-art scientific discussion. Openly accessible preprints play a crucial role in incorporating recent advancements, particularly in rapidly evolving fields such as aerosol–cloud interactions.

  The cited study was intended as an example to illustrate the impact of aerosols on deep convective cloud behavior, and another study could have been chosen to support this point. However, since the paper has now been accepted in Atmospheric Chemistry and Physics (ACP), we will update the citation accordingly. In the future, we will ensure that preprints are explicitly marked as such in the text.

**Reviewer Comment 1.2** — How did the authors determine that all the dust came from the Sahara? Relevant weather pattern analysis is required.

**Reply 1.2**: It is well established that during spring and summer, the Saharan desert is the primary source of mineral dust transported into Europe, as demonstrated in numerous studies analyzing atmospheric dust transport and associated weather patterns like you suggested (Moulin et al., 1997; Varga, 2020; Brunner et al., 2021). We therefore believe that the terminology "Saharan dust" is justified. On L21 we outline our reasoning regarding this comment, highlighting the main atmospheric processes driving northward Saharan dust transport: "...are the predominant source of atmospheric dust loads in the region over the past 40 years (Varga, 2020; Brunner et al., 2021).These dust plumes contribute significantly to European aerosol concentrations, affecting weather patterns and precipitation (Rodríguez et al., 2001; Masson et al., 2010). Subtropical anticyclones shifting to higher latitudes and amplified Rossby waves are associated with extreme Saharan dust events (Rodríguez and López-Darias, 2024)." We do not consider additional atmospheric transport analyses (e.g., trajectory modeling) necessary or within the scope of this study, as the dust's origin primarily serves to define our title.

**Reviewer Comment 1.3** — Lines 81-84: The author declared that they mainly focus on to investigate the influence of dust aerosol on hail occurrence, but only local days with lightning were included. Can it be understood that hail and lightning occur simultaneously? It should be described in more detail to make it easier for readers to understand.

**Reply 1.3**: We thank the reviewer for pointing out the need for clarification. Indeed, hail is an atmospheric phenomenon that always coincides with lightning, but the reverse is not true — thunderstorms can produce lightning without generating hail. Lightning forms in convective storms due to interactions between ice, hail, and supercooled water particles. As these particles collide, they transfer electrical charges, leading to a negatively charged cloud base and a positively charged top. This charge buildup generates an electric field, and once it becomes strong enough, lightning occurs. However, hail is not always observed at the surface during thunderstorms, as ice, graupel, or hailstones can melt before reaching the ground (if the melting level is too high or the hailstones are too weak).

To make this point more clear we adjusted L82: "...only local days with lightning (hereafter coined thunderstorm days) were included in the analysis ..."

**Reviewer Comment 1.4** — When the availability of the OPERA data is less than 100% in a 1°×1° grid, how is the hail area fraction calculated?

**Reply 1.4**: In that case, the fraction of the remaining area is computed.

**Reviewer Comment 1.5** — Lines 106-109: Do you mean that there are only 140 grid-points are available using EURADHAIL for determine hail events? I found that it conflicts with Figure 2.

**Reply 1.5**: Thank you for pointing out this conflict, the correct number of gridpoints is in fact 324. We've changed this accordingly in the revised version of the manuscript.

**Reviewer Comment 1.6** — Lines 114-17: This sentences "POH is an empirical hail detection algorithm estimating ground-level hail probability $(0 - 100\%)$ based on the vertical distance between

the 45 dBZ echo top height and the freezing level height, following Waldvogel et al. (1979). This approach is more accurate in capturing hail events than EURADHAIL, since it does not include the freezing level." confused me. Freezing level height is used to judge hail events, why does the author claim that this algorithm is more accurate than EURADHAIL because it does not include freezing level?

**Reply 1.6**: We appreciate the reviewer's comment and recognize the need for clarification. Our statement was meant to highlight the difference between POH and EURADHAIL: while POH explicitly incorporates the freezing level in its calculation, EURADHAIL does not. The freezing level height is crucial for identifying hail, as it determines the altitude at which hailstones begin to melt on their way to the ground. In the POH algorithm, the vertical distance between the freezing level and the 45 dBZ echo top serves as a proxy for the hail growth zone. A greater vertical extent allows more time for supercooled liquid water droplets to grow into hailstones within the storm's updraft. In contrast, EURADHAIL identifies hail solely based on exceeding a radar reflectivity threshold of 53 dBZ. Changes detailed in Reply 2.4 make the sentence on L117 more clear.

**Reviewer Comment 1.7** — Many of the labels on the horizontal and vertical axes of the figures are incomplete and need to be carefully modified.

**Reply 1.7**: Thank you for mentioning this, there was an issue during preprint publication with the embedded fonts, this will be addressed in the posting of the revised manuscript. See also AC1 on the ACP discussion page.

**Reviewer Comment 1.8** — Line 145: Thunderstorm day or hail day, which one is right? The same question in the title of Figure 1 and Figure 4.

**Reply 1.8**: Thank you for pointing out this ambiguity, we've specified the descriptor on L145, the two figure captions were correct.

**Reviewer Comment 1.9** — More detailed description about the Q should be added in Figure 1. In addition, how to calculated the fraction of hail days in Figure 3?

**Reply 1.9**: We've added additional information to the caption of Fig. 1: "($Q_{.3}$ and $Q_{.7}$ respectively)".

We consider the caption of Fig. 3 sufficiently self-explanatory and see no need for further elaboration in the main text.

**Reviewer Comment 1.10** — In figure 1, the mass of dust concentration is only divided into 2 groups. If the dust mass concentration is divided into three groups, does the maximum hail area fraction change with the dust mass concentration group as described in the manuscript?

**Reply 1.10**: The intermediate dust concentrations falls between the two distributions included in Fig. 1 (see Fig. R1). However, we don't see the added value of including this figure in the manuscript.

[Figure]

**Figure R1:** Equivalent to manuscript Fig. 1, with added intermediate dust loading group (green).

**Reviewer Comment 1.11** — Lines 156-158: Such analysis does not make sense, since aerosols of different scales co-exist in hail days.

**Reply 1.11**: This particular analysis is not intended to suggest that aerosol size modes act in isolation during hail events. Instead, it serves to test whether the relationship between dust and hail is sensitive to the dust size range (and dust size distribution) considered. No changes were made to the manuscript regarding this comment.

**Reviewer Comment 1.12** — Lines 180-184 and Lines 190-192: How does the author determine the optimal number of clustering centers?

**Reply 1.12**: As mentioned on L184, 11 clusters resulted in the optimum Silhouette score. We've rephrased for clarity: "In the selected target region, $n = 254$ lightning days were assigned to 11 clusters, the number of clusters was chosen based on the maximum Silhouette score (0.25) (Rousseeuw, 1987)."

**Reviewer Comment 1.13** — Lines 225-226: "availability" should be "variable". Why are different moisture variable used in LRM and GAM models.

**Reply 1.13**: Thank you for this comment. We have changed the wording in line 225: "The variables capturing atmospheric moisture availability are the 2 m dewpoint temperature (Td2m) in the LRM and the mean relative humidity across 850–500 hPa (RHmid) in the GAM." The choice of different moisture predictors is based on the distinct nature of the statistical models. Logistic regression (LRM) relies on a linear combination of predictors, meaning it struggles to accurately capture nonlinear relationships. In contrast, the generalized additive model (GAM) uses a linear combination of **functions** of predictors, allowing for more flexibility in modeling complex dependencies. For atmospheric moisture, there is a threshold beyond which additional

moisture no longer increases hail probability, as excessive moisture can burden storm updrafts by its "load", reducing the buoyancy. Residual analysis showed that the LRM struggled with this nonlinear relationship when using certain moisture variables. The GAM, with its greater flexibility, better captures this effect, improving overall performance (see Table 1). For this reason, selecting the most fitting predictors for each model is essential, particularly in an ingredients-based modeling approach.

**Reviewer Comment 1.14** — Lines 226-229: This sentence confused me. Which variable is most important for hail event prediction, dust loading or CAPE?

**Reply 1.14**: We appreciate the reviewer's comment and recognize the need for clarification. CAPE, representing atmospheric instability, was consistently the most important predictor for hail occurrence in both models. Dust and moisture followed in importance, with their ranking varying depending on the model type. In the LRM, dust was the second most important predictor, while in the GAM, moisture ranked slightly higher than dust. However, in both models, dust was always among the top three predictors. Since our goal is to highlight the "**relevance** of dust-related predictors in statistical hail modeling" (L229), rather than their exact **ranking**, the distinction between second and third place is not critical—especially given that this ranking can vary depending on model type, data resolution, covariates and parameter tuning.

We have adjusted L228f: "CAPE was the most important predictor in both models, followed by dust and/or moisture in second and third place, depending on the model. This was determined from z-values in the LRM and Chi-squared/SHAP values in the GAM."

**Reviewer 2**

**Summary**

This is a very interesting and relevant study investigating the effect of dust concentration on hail using observations. The methodology is sound, although the authors could be a bit more careful in the interpretation of some of the results and discuss the uncertainties more (see specific comments). I like that the manuscript is kept short and precise. However, in some parts a bit more detail might be necessary in both the literature background and the analysis (see specific comments). Most of my comments are minor and I don't see any reason to stop publication, but I strongly recommend to work on the following aspects.

**Reply**: Thank you for the constructive and encouraging evaluation. We appreciate your positive assessment of the study's relevance, methodological soundness, and concise presentation. We acknowledge the need for more cautious interpretation and a clearer discussion of uncertainties, and we have addressed these in the revised manuscript as per your specific comments.

**Specific comments**

**Reviewer Comment 2.1** — Lines 27-67: Perhaps the authors are more familiar with the topic of aerosol-cloud interactions (focused on hail) than me, but isn't this topic much less clear than portrayed here? To my knowledge, there are some contradicting results in the literature (see e.g., the sections on aerosol effects in the reviews of Allen et al. (2020) and Raupach et al. (2021)) while herein the different physical processes are portrayed as clear picture in just a few short sentences in each paragraph. I'm no expert on this topic but I think some more context might be good on what processes are still uncertain.

**Reply 2.1**: To address the comment, we added two sentences (L67) referencing the reviews by Allen et al. (2020) and Raupach et al. (2021), highlighting the persisting contradictions and uncertainties in the field. This addition clarifies that while many studies support specific aerosol effects, the overall understanding of aerosol–cloud–precipitation interactions remains incomplete and context-dependent: "However, the overall picture remains complex and partially inconsistent. Comprehensive reviews by Allen et al. (2020) and Raupach et al. (2021) emphasize that aerosol–cloud–precipitation interactions are not yet fully understood, with diverging findings depending on storm type, model setup, and environmental conditions."

**Reviewer Comment 2.2** — Most Figures are missing labels. I saw that you added the corrected Figures in your reply to reviewer 1, so this seems resolved?

**Reply 2.2**: This has been resolved (see Reply 1.7).

**Reviewer Comment 2.3** — Line 39: Do you mean here that beside the change in stability, thermal convection is reduced, which inhibits convection initiation? If yes, I suggest writing "thermal

convection" or "boundary layer thermals" instead of just "convection". Furthermore, this "negative" impact of aerosols on CI has also been discussed as a possibly important factor in strong Saharan dust scenarios over Europe (Seifert et al., 2023; Fischer et al., 2024). I think these negative effects could be discussed a bit more in the manuscript since this fits the decline in hail occurence with high dust concentrations in your study (see also comment 14).

**Reply 2.3**: Thank you for pointing this out, we've changed it to "thermal convection". Furthermore, we've made the following insertion on L41: "This atmospheric heating can also inhibit convection initiation under certain conditions, particularly in strong Saharan dust scenarios, where increased atmospheric stability has been linked to a decline in convective activity and hail occurrence (Seifert et al., 2023; Fischer et al., 2024).".

**Reviewer Comment 2.4** — Line 117: Better write "the latter" instead of "it" to make clear that you are referring to EURADHAIL (right?).

**Reply 2.4**: Thank you, we've implemented this change as you've suggested.

**Reviewer Comment 2.5** — Fig 3: I'm not sure I understand this fig or how you interpret it. There is a peak, but the decrease in hail fraction with lower dust concentrations is very small. So doesn't this show that hail potential doesn't change much at lower concentrations?
    Perhaps related to this, why is the plot cut off at low concentrations?

**Reply 2.5**: We acknowledge the reviewer's observation that the decrease in hail fraction at low dust concentrations appears small in Fig. 3. However, this figure specifically shows the fraction of local thunderstorm days that meet the hail-day criterion ($\geq 1\%$ area coverage per $1 \times 1°$ grid box, as defined in Sect. 2.2.1). For this threshold, the curve is indeed relatively flat at lower dust concentrations, suggesting that low concentrations are not strongly limiting for localized hail occurrence.
    However, as shown in Fig. 1 and discussed in the corresponding section, increasing the area fraction threshold reveals a much clearer sensitivity to dust: more spatially extensive hail events become significantly less likely at low dust concentrations. Thus, while Fig. 3 may suggest weak sensitivity at the low end for localized events, the broader context provided by Fig. 1 confirms that low dust concentrations do limit the occurrence of widespread hail events.
    No changes to the manuscript were made, as the current figures and text already convey this nuance.
    In response to your related comment: only 2% of days exist in the climatology that are beyond the x-axis limit (see new Fig. A1), the limit was chosen to maximize the readability of the figure.

**Reviewer Comment 2.6** — To interpret Fig. 3 and more generally the context of dust concentrations I think it would be helpful to add a Figure showing a histogram of dust concentrations underlying your analysis. In other words, how frequent are concentrations e.g., of $> 20$ mg/m2. You only briefly mention some context on the underlying distribution in line 164.

**Reply 2.6**: We've included an additional figure in the appendix (Fig. A1) and referred to it where appropriate.

**Reviewer Comment 2.7** — Section 3.2: I like that you looked into the possible link to weather patterns and I mostly agree with your conclusion. However, I think it is still possible that even within one general synoptic setting higher dust could not be causally linked to more hail but just be correlated with larger-scale processes important for hail formation. For example, it is known that steep lapse-rates are important for hail, which is often found when an elevated mixed layer is advected from the Iberian Peninsula or Africa, areas which are major sources of dust (Schultz et al., 2025). It's hard to say how well your clustering approach is capturing complex processes like this. Is it possible that even within one of your clusters, there might be days in which the flow supports EML and dust formation and in others it doesn't? Then dust would only be correlated to hail because of the increased lapse-rates. Considering mentioning this possibility.

**Reply 2.7**: We added a sentence (L199) to acknowledge that within one synoptic cluster, thermodynamic differences — such as the presence of elevated mixed layers advected from dust source regions — may still exist and confound the dust-hail relationship (Schultz et al., 2025): "Even within a given synoptic cluster, days may differ in thermodynamic structure, e.g., presence of elevated mixed layers advected from dust source regions (Schultz et al., 2025). This could lead to a spurious dust-hail link driven by lapse-rate changes rather than dust itself.".

**Reviewer Comment 2.8** — Line 187: Would it be worth showing the general flow pattern for your clusters? I think this could be interesting because it would show in what synoptic scenarios dust is not important (0.81?).

**Reply 2.8**: We appreciate the suggestion and have included the corresponding composite flow pattern for all clusters in the response (see Fig. R2). However, we do not consider this figure essential for the manuscript, as the added interpretative value is limited and does not directly support the core conclusions. For example, the cluster with a dust mass factor of 0.81 (Cluster 3) does not exhibit a clearly distinct synoptic pattern that would allow robust generalization about dust irrelevance under specific scenarios.

**Reviewer Comment 2.9** — Line 189: The coordinate ranges don't mean much to me. Consider adding a map for target regions (e.g., add to fig 2) or at least say more about why they were chosen.
    Also, remove "a" before "different".

**Reply 2.9**: Thank you for bringing up this usability issue; we've added respective geographical descrip tors to the target region specifications (eastern Austria, Croatia, and central France). Furthermore, on L183 we've specified the region as "...centered on the target region north of the Alps (7°E to 12°E & 47°N 49°N)."
    Also, we've removed the erroneous "a".

**Reviewer Comment 2.10** — Line 213: Unclear why "(1)" is added here.

**Reply 2.10**: Thank you for spotting this, it was supposed to be "(Tab 1)". We've changed this accordingly.

[Figure]

**Figure R2:** Mean 15 UTC geopotential height $Z$ at $p = 500\,\text{hPa}$ height of the 11 clusters in a $14° \times 20°$ box (coarsened to $7 \times 7$ pixels) centered on the target region north of the Alps (7°E to 12°E & 47°N 49°N) for $n = 254$ lightning days.

**Reviewer Comment 2.11** — Line 239: I don't follow your argument here. Is the dust concentration so heterogeneous from one grid point to the next that averaging over such a small regions would change the threshold so significantly?

**Reply 2.11**: Thank you for the comment. Some degree of local variability in dust concentration is indeed expected. Moreover, the analysis in Sect. 3 is performed across the entire EURADHAIL domain, while the value cited here is based on a smaller subregion in northern Switzerland. This regional focus, combined with local variability, can plausibly lead to a lower dust threshold in this specific context. No changes were made to the manuscript regarding this comment.

**Reviewer Comment 2.12** — Section 4: I think the statistical modeling is a good idea to highlight the relevance of dust. However, one may criticize that adding and almost any additional predictor ise expected to enhance model performance slightly, so comparing the slight improvements of the model with and without dust could be misleading, no?

**Reply 2.12**: The reviewer's general point is correct: adding predictors can often lead to marginal performance gains, especially in models that are not properly tuned or regularized. However, in our modeling process, we tested many additional predictors (including several thermodynamic and kinematic variables), and in most cases, these did not meaningfully improve model performance or yielded statistically insignificant coefficients. The improvements observed when including dust predictors are therefore not an automatic outcome of variable inclusion but stand out as non-trivial and robust.

This is particularly relevant given that our models were well-tuned and subject to cross-validation and out-of-sample testing. The dust variable consistently improved key metrics (e.g., +12% CSI, +6% explained variance in GAMs), indicating added predictive value beyond statistical noise. That said, we do not claim that these gains are universally transferable to all models or setups. The observed improvements depend on the specific predictors, model architecture, and target metric.

Our core argument is not that dust loading universally improves every hail model, but that it is a rarely tested predictor with demonstrated potential. Given its physical plausibility and observed contribution in our context, future studies may benefit from considering it explicitly, while remaining cautious of model-specific constraints. The section is therefore defensible as written.

**Reviewer Comment 2.13** — Relatedly, it's interesting and supporting your conclusions that dust is more important than other predictors like wind shear, but this opens another question as wind shear is known to be important for hail storms. Could you elaborate on this? Are wind shear and dust highly-correlated so that the model only needs one? Or perhaps wind shear is not so important because the model is trained on hail coverage and not hail size?

**Reply 2.13**: This point is acknowledged in the manuscript through two mechanisms. First, it is explicitly stated that CAPE is the dominant predictor, while the importance of dust versus moisture and shear depends on model type. Second, the manuscript focuses on hail occurrence rather than severity, which likely affects the relative importance of predictors.

Wind shear is indeed a known control for hail intensity, especially large hail, but its role in occurrence is secondary to instability and moisture availability. The models are trained on binary hail day classification based on radar-derived coverage, not on metrics of hail size or storm organization where shear would play a more central role. Thus, the observed lower importance of wind shear does not contradict established understanding but reflects the specific prediction target.

Additionally, multicollinearity was assessed and is low; dust and wind shear are not highly correlated. Their effects are distinguishable, and dust remains a significant predictor even in the presence of shear. This supports the conclusion that dust adds independent predictive value.

No changes needed. The existing text and model design implicitly address this comment.

**Reviewer Comment 2.14** — In Fig. 5, it looks like the added value of dust as predictor is mostly from its negative effects (see comment 3) yet the hail-enhancing influence is emphasized a lot more in your text (take for example the manuscript title and abstract). Consider writing about both effects in a more balanced way.

**Reply 2.14**: We agree that Fig. 5 shows both a hail-enhancing and hail-suppressing influence of dust. However, the suppressing effect becomes relevant only at very high dust concentrations, which are rare. This is indicated by the short black vertical lines in Fig. 5, representing the relative frequency of the corresponding dust values (see also Fig. A1). The added predictive value of dust is thus mainly due to its frequent enhancement effect in the intermediate range (up to $40 \,\mathrm{mg\,m^{-2}}$), while the suppressive effect at higher concentrations affects only a small number of events.

The Abstract already clearly mentions negative effects (L8): "Peak hail activity occurs at $38 \,\mathrm{mg\,m^{-2}}$ or a dust optical depth of 0.033, suggesting enhanced cloud and ice nucleation. Above this range, hail frequency declines, likely due to microphysical or radiative constraints."

Nevertheless, to reflect both effects more clearly, we have revised the last sentence of the Abstract (L14) to present a more balanced view: "These findings identify Saharan dust as a key modulator of European hail activity, exerting both enhancing and inhibiting effects depending on dust concentration and the definition of hail events."

**Reviewer Comment 2.15** — Your study is mostly based on EURADHAIL and POH as truth for hail. Both heavily rely on radar reflectivity, which can also be high in the presence of strong liquid precipitation. So, could the link you find between dust and hail be at least partially a result of the influence of dust on heavy precip (Zhu et al. 2024)? In other words, even with hail staying equal, an increase in precipitation intensity would result in an increase in reflectivity max and reflectivity area and therefore falsely show an impact of dust in your study. I agree that hail is likely dominating these reflectivity-based parameters, but the question is how big the impact of intensifying precip is. If you agree, this uncertainty should be discussed.

**Reply 2.15**: This is certainly a concern, however, we also find a robust effect when using crowd-sourced hail reports in place of radar retrievals (as discussed in Sect. 3.1). Furthermore, POH also incorporates the vertical extent of the thunderstorm into it's estimation of hail and not just reflectivity (see also Reply 1.6). We've added the following sentence to L297: "Furthermore, the potential bias due to dust-enhanced precipitation intensity (e.g., Zhu et al., 2024) affecting radar reflectivity cannot be fully excluded, though the consistent signal observed in crowd-sourced hail reports (Sect. 3.1) supports a genuine hail-related effect of the findings presented throughout this study."

**Reviewer Comment 2.16** — Related to comment 1, the strong influence of dust on hail you suggest opens the question why other regions of the world which are less directly influenced by major dust sources like the Sahara are still having intense hailstorms hail (e.g., Northern US or South America). Any thoughts?

**Reply 2.16**: We acknowledge this important point. The observation that intense hailstorms also occur in regions with minimal direct dust influence indicates that dust is not a necessary condition for severe hail formation. Instead, our findings suggest that in regions like Europe, where dust is intermittently abundant, its presence may act as a catalyst enhancing hailstorm development under otherwise conducive conditions. We've added an outlook regarding your comment to the last section (L300): "Future studies should assess whether similar dust–hail relationships hold in other

hail-prone regions globally, particularly those with different or limited aerosol sources, such as the central United States or South America."

**Reviewer Comment 2.17** — I also agree with comment 2 of reviewer 1: How do you know that Saharan dust is dominating? Perhaps you could elaborate a bit on the robustness of the CAMS data and why Saharan dust is most likely?

**Reply 2.17**: We argue that the sentence on L20 and the following adequately addresses this (Saharan dust events, characterized by the transport of mineral dust from the Sahara Desert across the Mediterranean into Europe, are the predominant source of atmospheric dust loads in the region over the past 40 years (Varga, 2020; Brunner et al., 2021).). See also Reply 1.2, no changes were made to the manuscript regarding this comment.

**References**

Allen, J. T., I. M. Giammanco, M. R. Kumjian, H. Jurgen Punge, Q. Zhang, P. Groenemeijer, M. Kunz, and K. Ortega, 2020: Understanding hail in the earth system. *Rev. Geophys.*, **58 (1)**, DOI: 10.1029/2019RG000665.

Brunner, C., B. T. Brem, M. Collaud Coen, F. Conen, M. Hervo, S. Henne, M. Steinbacher, M. Gysel-Beer, and Z. A. Kanji, 2021: The contribution of Saharan dust to the ice-nucleating particle concentrations at the High Altitude Station Jungfraujoch (3580 m a.s.l.), Switzerland. *Atmos. Chem. Phys.*, **21 (23)**, 18029–18053, DOI: 10.5194/acp-21-18029-2021.

Fischer, J., P. Groenemeijer, A. Holzer, M. Feldmann, K. Schröer, F. Battaglioli, L. Schielicke, T. Púčik, C. Gatzen, B. Antonescu, and the TIM Partners, 2024: Invited perspectives: Thunderstorm intensification from mountains to plains. *EGUsphere*, 1–41, DOI: 10.5194/egusphere-2024-2798.

Masson, O., D. Piga, R. Gurriaran, and D. d'Amico, 2010: Impact of an exceptional Saharan dust outbreak in France: PM10 and artificial radionuclides concentrations in air and in dust deposit. *Atmos. Environ.*, **44 (20)**, 2478–2486, DOI: 10.1016/j.atmosenv.2010.03.004.

Moulin, C., C. E. Lambert, F. Dulac, and U. Dayan, 1997: Control of atmospheric export of dust from North Africa by the North Atlantic Oscillation. *Nature*, **387 (6634)**, 691–694, DOI: 10.1038/42679.

Raupach, T. H., O. Martius, J. T. Allen, M. Kunz, S. Lasher-Trapp, S. Mohr, K. L. Rasmussen, R. J. Trapp, and Q. Zhang, 2021: The effects of climate change on hailstorms. *Nat. Rev. Earth Environ.*, **2 (3)**, 213–226, DOI: 10.1038/s43017-020-00133-9.

Rodríguez, S., X. Querol, A. Alastuey, G. Kallos, and O. Kakaliagou, 2001: Saharan dust contributions to PM10 and TSP levels in southern and eastern Spain. *Atmos. Environ.*, **35 (14)**, 2433–2447, DOI: 10.1016/S1352-2310(00)00496-9.

Rodríguez, S. and J. López-Darias, 2024: Extreme Saharan dust events expand northward over the Atlantic and Europe, prompting record-breaking $PM_{10 \text{ and } PM_{2.5} \text{ episodes}}$. *Atmos. Chem. Phys.*, **24 (20)**, 12031–12053, DOI: 10.5194/acp-24-12031-2024.

Rousseeuw, P. J., 1987: Silhouettes: a graphical aid to the interpretation and validation of cluster analysis. *J. Comput. Appl. Math.*, **20**, 53–65, DOI: `10.1016/0377-0427(87)90125-7`.

Schultz, D. M., M. V. Young, and D. J. Kirshbaum, 2025: The Spanish Plume Elevated Mixed Layer: A Review of Its Use and Misuse within the Scientific Literature. *Monthly Weather Review*, **153 (5)**, 737–761, DOI: `10.1175/MWR-D-24-0139.1`.

Seifert, A., V. Bachmann, F. Filipitsch, J. Förstner, C. M. Grams, G. A. Hoshyaripour, J. Quinting, A. Rohde, H. Vogel, A. Wagner, and B. Vogel, 2023: Aerosol–cloud–radiation interaction during Saharan dust episodes: the dusty cirrus puzzle. *Atmospheric Chemistry and Physics*, **23 (11)**, 6409–6430, DOI: `10.5194/acp-23-6409-2023`.

Varga, G., 2020: Changing nature of Saharan dust deposition in the Carpathian Basin (Central Europe): 40 years of identified North African dust events (1979–2018). *Environ. Int.*, **139**, 105712, DOI: `10.1016/j.envint.2020.105712`.

Zhu, L., Y. Wang, D. Chavas, M. Johncox, and Y. L. Yung, 2024: Leading role of Saharan dust on tropical cyclone rainfall in the Atlantic Basin. *Sci. Adv.*, **10 (30)**, eadn6106, DOI: `10.1126/sciadv.adn6106`.

---

## Author Response (AR2)

**Round two: Author's comments for paper egusphere-2024-3924**

**Saharan dust linked to European hail events**

by Killian P. Brennan and Lena Wilhelm

July 21, 2025

We thank Reviewer 1 for their continued engagement with our work. Several comments raised in this second round overlap or are in some instances identical with points addressed during the initial review. Where appropriate, we have reiterated and clarified our responses and made further adjustments to improve clarity and consistency in the manuscript. We thank Reviewer 2 for recommending acceptance of the manuscript as is. We trust that the current version satisfactorily resolves all outstanding points.
* * *
**Reviewer 1**

**Overview**

Dust aerosols play an important role in the formation of clouds, especially in the ice phase processes within the clouds. This study discusses the impact of Saharan dust aerosols on hail in Europe, which has significant scientific implications.

**Reply**: We thank the reviewer for acknowledging the scientific relevance of our study and for providing additional comments aimed at improving clarity and methodological transparency. Several comments in this round appear to reiterate points already raised and addressed in the first review round. While this may reflect a misunderstanding or an oversight in consulting our previous responses, we have nevertheless restated and, where helpful, clarified our replies to ensure that our rationale is clear and traceable. In addition, we have implemented minor edits to improve consistency and transparency in the manuscript. We hope these clarifications resolve the remaining concerns.

**Specific comments**

**Reviewer Comment 1.1** — How did the authors determine that all the dust came from the Sahara? Relevant weather pattern analysis is required.

**Reply 1.1**: This comment is identical to Reviewer Comment 1.2 in the first round, where we respond with: "It is well established that during spring and summer, the Saharan desert is the

primary source of mineral dust transported into Europe, as demonstrated in numerous studies analyzing atmospheric dust transport and associated weather patterns like you suggested (Moulin et al., 1997; Varga, 2020; Brunner et al., 2021). We therefore believe that the terminology "Saharan dust" is justified. On L21 we outline our reasoning regarding this comment, highlighting the main atmospheric processes driving northward Saharan dust transport: "...are the predominant source of atmospheric dust loads in the region over the past 40 years (Varga, 2020; Brunner et al., 2021).These dust plumes contribute significantly to European aerosol concentrations, affecting weather patterns and precipitation (Rodríguez et al., 2001; Masson et al., 2010). Subtropical anticyclones shifting to higher latitudes and amplified Rossby waves are associated with extreme Saharan dust events (Rodríguez and López-Darias, 2024)." We do not consider additional atmospheric transport analyses (e.g., trajectory modeling) necessary or within the scope of this study, as the dust's origin primarily serves to define our title."

**Reviewer Comment 1.2** — Is there a clear connection between lightning and hail? Why does the research on the impact of sandstorms on the frequency of hail occurrence only focus on days when lightning is present?

**Reply 1.2**: In our eyes, Reply 1.3 in the first round adequately addresses this comment: "Indeed, hail is an atmospheric phenomenon that always coincides with lightning, but the reverse is not true — thunderstorms can produce lightning without generating hail. Lightning forms in convective storms due to interactions between ice, hail, and supercooled water particles." See also justification on L86f on why we only look at local days with lightning: "In order to investigate the effect of altered dust loads on hail occurrence and not thunderstorm occurrence in general, only local days with lightning (hereafter coined thunderstorm days) were included in the analysis performed in this study."

**Reviewer Comment 1.3** — When the coverage of OPERA data is less than 100% in grid-point area, will the hail area fraction affect the statistical results? The author should give a more detail analysis.

**Reply 1.3**: We thank the reviewer for the follow-up. As stated in our response to Comment 1.4 of the previous round, the hail area fraction is computed relative to the area with available OPERA coverage. This approach ensures internal consistency across all grid cells, regardless of coverage percentage.

**Reviewer Comment 1.4** — thunderstorm day, hail day, non-thunderstorm day, and non-hail day should be unified, it confuses the reader.

**Reply 1.4**: We thank the reviewer for this observation. We have reviewed the terminology and confirmed that "non-thunderstorm day" was never used. To improve clarity, we now consistently use "non-hail" throughout the manuscript, replacing earlier instances of "non-hail, thunderstorm days", the terminology is specified in a footnote on L153. Furthermore, "thunderstorm days" is now used consistently in place of "lightning days".

**Reviewer Comment 1.5** — Lines 233-234: Why are different moisture variable used in LRM and GAM models.

**Reply 1.5**: Thank you for this comment. This question is identical to Reviewer Comment 1.13 from the first review round. As previously explained, in an ingredients-based modeling approach, the aim is to select the best combination of predictors for each model type to achieve optimal performance, rather than to use an identical set of variables across all models. Logistic regression (LRM) and generalized additive models (GAMs) differ in their statistical structure. LRM models linear relationships between predictors and the response, while GAMs allow for nonlinear (non-parametric) relationships through smooth functions of the predictors. This distinction is particularly relevant for atmospheric moisture, where the relationship with hail probability is not strictly linear. After a certain point, additional moisture does not continue to increase hail probability indefinitely, due to physical constraints. Very large moisture loading can dampen the updraft strength and reduce boyancy. Residual analyses indicated that the LRM struggled to represent this effect when using certain moisture variables, whereas the GAM, due to its flexibility, performed better when using relative humidity as a predictor (see Table 1).

We believe the original explanation already provides a thorough physical and statistical justification for the use of different moisture variables. There is no further reasoning beyond the goal of model-specific optimization.

To make this clearer in the manuscript, we propose adding the following sentence after lines 233–234: "Because of their differing model structures, GAMs benefited from using mid-level relative humidity, which provided better predictive performance than $2\,\mathrm{m}$ dewpoint temperature due to better capturing the complex role of moisture availability in hail formation."

**Reviewer 2**

**Summary**

The authors thoroughly addressed my comments. My impression when reading the responses was that the authors were annoyed by some of my comments, which were only meant as suggestions for improvement (apologies if they seemed overly criticizing). So to save everyone's time, I won't press these points more. The article is excellent overall and I didn't notice anything else.

**Reply**: We thank the reviewer for their thoughtful feedback and kind words. We apologize if any of our responses came across as dismissive or annoyed — that was not our intention. We greatly appreciated the reviewer's suggestions, which helped us improve the manuscript.

**References**

Brunner, C., B. T. Brem, M. Collaud Coen, F. Conen, M. Hervo, S. Henne, M. Steinbacher, M. Gysel-Beer, and Z. A. Kanji, 2021: The contribution of Saharan dust to the ice-nucleating particle concentrations at the High Altitude Station Jungfraujoch (3580 m a.s.l.), Switzerland. *Atmos. Chem. Phys.*, **21 (23)**, 18029–18053, DOI: `10.5194/acp-21-18029-2021`.

Masson, O., D. Piga, R. Gurriaran, and D. d'Amico, 2010: Impact of an exceptional Saharan dust outbreak in France: PM10 and artificial radionuclides concentrations in air and in dust deposit. *Atmos. Environ.*, **44 (20)**, 2478–2486, DOI: `10.1016/j.atmosenv.2010.03.004`.

Moulin, C., C. E. Lambert, F. Dulac, and U. Dayan, 1997: Control of atmospheric export of dust from North Africa by the North Atlantic Oscillation. *Nature*, **387 (6634)**, 691–694, DOI: `10.1038/42679`.

Rodríguez, S., X. Querol, A. Alastuey, G. Kallos, and O. Kakaliagou, 2001: Saharan dust contributions to PM10 and TSP levels in southern and eastern Spain. *Atmos. Environ.*, **35 (14)**, 2433–2447, DOI: `10.1016/S1352-2310(00)00496-9`.

Rodríguez, S. and J. López-Darias, 2024: Extreme Saharan dust events expand northward over the Atlantic and Europe, prompting record-breaking $PM_{10 \text{ and } PM_{2.5} \text{ episodes}}$. *Atmos. Chem. Phys.*, **24 (20)**, 12031–12053, DOI: `10.5194/acp-24-12031-2024`.

Varga, G., 2020: Changing nature of Saharan dust deposition in the Carpathian Basin (Central Europe): 40 years of identified North African dust events (1979–2018). *Environ. Int.*, **139**, 105712, DOI: `10.1016/j.envint.2020.105712`.